# PROBABILITY CALIBRATION FOR KNOWLEDGE GRAPH EMBEDDING MODELS

**Pedro Tabacof, Luca Costabello**
Accenture Labs
Dublin, Ireland
{pedro.tabacof, luca.costabello}@accenture.com

## ABSTRACT

Knowledge graph embedding research has overlooked the problem of probability calibration. We show popular embedding models are indeed uncalibrated. That means probability estimates associated to predicted triples are unreliable. We present a novel method to calibrate a model when ground truth negatives are not available, which is the usual case in knowledge graphs. We propose to use Platt scaling and isotonic regression alongside our method. Experiments on three datasets with ground truth negatives show our contribution leads to well calibrated models when compared to the gold standard of using negatives. We get significantly better results than the uncalibrated models from all calibration methods. We show isotonic regression offers the best the performance overall, not without trade-offs. We also show that calibrated models reach state-of-the-art accuracy without the need to define relation-specific decision thresholds.

## 1 INTRODUCTION

Knowledge graph embedding models are neural architectures that learn vector representations (i.e. embeddings) of nodes and edges of a knowledge graph. Such knowledge graph embeddings have applications in knowledge graph completion, knowledge discovery, entity resolution, and link-based clustering, just to cite a few (Nickel et al., 2016a).

Despite burgeoning research, the problem of calibrating such models has been overlooked, and existing knowledge graph embedding models do not offer any guarantee on the probability estimates they assign to predicted facts. Probability calibration is important whenever you need the predictions to make probabilistic sense, i.e., if the model predicts a fact is true with 80% confidence, it should to be correct 80% of the times. Prior art suggests to use a sigmoid layer to turn logits returned by models into probabilities (Nickel et al., 2016a) (also called the expit transform), but we show that this provides poor calibration. Figure 1 shows reliability diagrams for off-the-shelf TransE and ComplEx. The identity function represents perfect calibration. Both models are miscalibrated: all TransE combinations in Figure 1a under-forecast the probabilities (i.e. probabilities are too small), whereas ComplEx under-forecasts or over-forecasts according to which loss is used (Figure1b).

Calibration is crucial in high-stakes scenarios such as drug-target discovery from biological networks, where end-users need trustworthy and interpretable decisions. Moreover, since probabilities are not calibrated, when classifying triples (i.e. facts) as true or false, users must define relation-specific thresholds, which can be awkward for graphs with a great number of relation types.

To the best of our knowledge, this is the first work to focus on calibration for knowledge embeddings. Our contribution is two-fold: First, we use Platt Scaling and isotonic regression to calibrate knowledge graph embedding models on datasets that include ground truth negatives. One peculiar feature of knowledge graphs is that they usually rely on the *open world assumption* (facts not present are not necessarily false, they are simply unknown). This makes calibration troublesome because of the lack of ground truth negatives. For this reason, our second and main contribution is a calibration heuristics that combines Platt-scaling or isotonic regression with synthetically generated negatives.

Experimental results show that we obtain better-calibrated models and that it is possible to calibrate knowledge graph embedding models even when ground truth negatives are not present. We

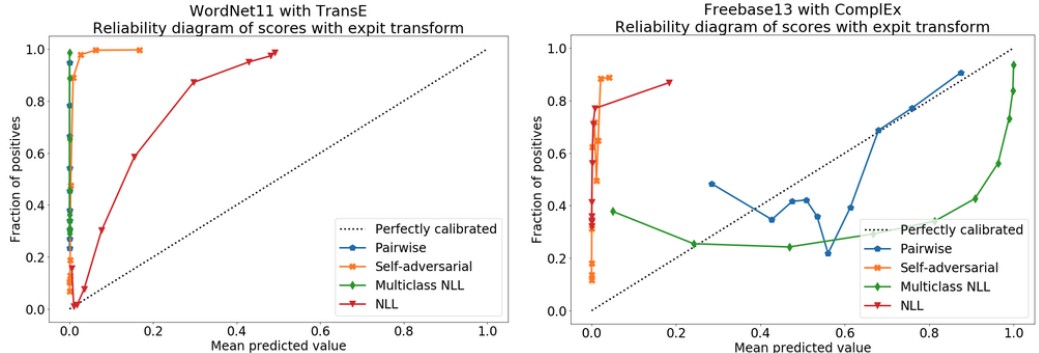

Figure 1: Reliability diagrams of uncalibrated models. Probabilities are generated by a logistic sigmoid layer. The larger the deviation from the diagonal, the more uncalibrated is the model. We present four different common loss functions used to train knowledge graph embedding models. (a) Uncalibrated TransE on WN11. (b) Uncalibrated ComplEx on FB13. Best viewed in colors.

also experiment with triple classification, and we show that calibrated models reach state-of-the-art accuracy without the need to define relation-specific decision thresholds.

## 2 RELATED WORK

A comprehensive survey of knowledge graph embedding models is out of the scope of this paper. Recent surveys such as (Nickel et al., 2016a) and (Cai et al., 2017) summarize recent literature.

TransE (Bordes et al., 2013) is the forerunner of distance-based methods, and spun a number of models commonly referred to as TransX. The intuition behind the symmetric bilinear-diagonal model DistMult (Yang et al., 2015) paved the way for its asymmetric evolutions in the complex space, RotatE (Sun et al., 2019) and ComplEx (Trouillon et al., 2016) (a generalization of which uses hypercomplex representations (Zhang et al., 2019)). HolE relies instead on circular correlation (Nickel et al., 2016b). The recent TorusE (Ebisu & Ichise, 2018) operates on a lie group and not in the Euclidean space. While the above models can be interpreted as multilayer perceptrons, others such as ConvE (Dettmers et al., 2018) or ConvKB (Nguyen et al., 2018) include convolutional layers. More recent works adopt capsule networks architectures (Nguyen et al., 2019). Adversarial learning is used by KBGAN (Cai & Wang, 2018), whereas attention mechanisms are instead used by (Nathani et al., 2019). Some models such as RESCAL (Nickel et al., 2011), TuckER (Balažević et al., 2019), and SimplE (Kazemi & Poole, 2018) rely on tensor decomposition techniques. More recently, ANALOGY adopts a differentiable version of analogical reasoning (Liu et al., 2017). In this paper we limit our analysis to four popular models: TransE, DistMult, ComplEx and HolE. They do not address the problem of assessing the reliability of predictions, leave aside calibrating probabilities.

Besides well-established techniques such as Platt scaling (Platt et al., 1999) and isotonic regression (Zadrozny & Elkan, 2002), recent interest in neural architectures calibration show that modern neural architectures are poorly calibrated and that calibration can be improved with novel methods. For example, (Guo et al., 2017) successfully proposes to use temperature scaling for calibrating modern neural networks in classification problems. On the same line, (Kuleshov et al., 2018) proposes a procedure based on Platt scaling to calibrate deep neural networks in regression problems.

The Knowledge Vault pipeline in (Dong et al., 2014) extracts triples from unstructured knowledge and is equipped with Platt scaling calibration, but this is not applied to knowledge graph embedding models. KG2E (He et al., 2015) proposes to use normally-distributed embeddings to account for the uncertainty, but their model does not provide the probability of a triple being true, so KG2E would also benefit from the output calibration we propose here. To the best of our knowledge, the only work that adopts probability calibration to knowledge graph embedding models is Krompaß & Tresp (2015). The authors propose to use ensembles in order to improve the results of knowledge graph embedding tasks. For that, they propose to calibrate the models with Platt scaling, so they

| | WN11 | FB13 | YAGO39K | FB15K-237 | WN18RR |
|---|---|---|---|---|---|
| Training | 112,581 | 316,232 | 354,996 | 272,115 | 86,835 |
| Validation | 5,218 | 11,816 | 18,682 | 17,535 | 3,034 |
| Test | 21,088 | 47,466 | 18,728 | 20,466 | 3,134 |
| Entities | 38,696 | 75,043 | 39,374 | 14,541 | 40,943 |
| Relations | 11 | 13 | 39 | 237 | 11 |

(a)

| Model | Scoring Function $f_m$ |
|---|---|
| TransE | $-\lvert\lvert \mathbf{e}_s + \mathbf{r}_p - \mathbf{e}_o \rvert\rvert_n$ |
| DistMult | $\langle \mathbf{e}_s, \mathbf{r}_p, \mathbf{e}_o \rangle$ |
| ComplEx | $Re(\langle \mathbf{e}_s, \mathbf{r}_p, \overline{\mathbf{e}_o} \rangle)$ |
| HolE | $\langle \mathbf{e}_s, \mathbf{r}_p \otimes \mathbf{e}_o \rangle$ |

(b)

Table 1: (a) Triple classification datasets used in experiments (left); link prediction datasets used for positive base rate experiments (right); (b) Scoring functions of models used in experiments.

operate on the same scale. No further details on the calibration procedure are provided. Besides, there is no explanation on how to handle the lack of negatives.

## 3 PRELIMINARIES

**Knowledge Graph.** Formally, a knowledge graph $\mathcal{G} = \{(s, p, o)\} \subseteq \mathcal{E} \times \mathcal{R} \times \mathcal{E}$ is a set of triples $t = (s, p, o)$, each including a subject $s \in \mathcal{E}$, a predicate $p \in \mathcal{R}$, and an object $o \in \mathcal{E}$. $\mathcal{E}$ and $\mathcal{R}$ are the sets of all entities and relation types of $\mathcal{G}$.

**Triple Classification**. Binary classification task where $\mathcal{G}$ (which includes only positive triples) is used as training set, and $\mathcal{T} = \{(s, p, o)\} \subseteq \mathcal{E} \times \mathcal{R} \times \mathcal{E}$ is a disjoint test set of *labeled* triples to classify. Note $\mathcal{T}$ includes positives and negatives. Since the learned models are not calibrated, multiple decision thresholds $\tau_i$ must be picked, where $0 < i < |\mathcal{R}|$, i.e. one for each relation type. This is done using a validation set (Bordes et al., 2013). Classification metrics apply (e.g. accuracy).

**Link Prediction.** Given a training set $\mathcal{G}$ that includes only positive triples, the goal is assigning a score $f(t) \in \mathbb{R}$ proportional to the likelihood that each *unlabeled* triple $t$ included in a held-out set $\mathcal{S}$ is true. Note $\mathcal{S}$ does not have ground truth positives or negatives. This task is cast as a learning to rank problem, and uses metrics such as mean rank (MR), mean reciprocal rank (MRR) or Hits@N.

**Knowledge Graph Embeddings.** Knowledge graph embedding models are neural architectures that encode concepts from a knowledge graph $\mathcal{G}$ (i.e. entities $\mathcal{E}$ and relation types $\mathcal{R}$) into low-dimensional, continuous vectors $\in \mathbb{R}^k$ (i.e, the embeddings). Embeddings are learned by training a neural architecture over $\mathcal{G}$. Although such architectures vary, the training phase always consists in minimizing a loss function $\mathcal{L}$ that includes a *scoring function* $f_m(t)$, i.e. a model-specific function that assigns a score to a triple $t = (s, p, o)$ (more precisely, the input of $f_m$ are the embeddings of the subject $\mathbf{e}_s$, the predicate $\mathbf{r}_p$, and the object $\mathbf{e}_o$). The goal of the optimization procedure is learning optimal embeddings, such that the scoring function $f_m$ assigns high scores to positive triples $t^+$ and low scores to triples unlikely to be true $t^-$. Existing models propose scoring functions that combine the embeddings $\mathbf{e}_s, \mathbf{r}_p, \mathbf{e}_o \in \mathbb{R}^k$ using different intuitions. Table 1b lists the scoring functions of the most common models. For example, the scoring function of TransE computes a similarity between the embedding of the subject $\mathbf{e}_s$ translated by the embedding of the predicate $\mathbf{e}_p$ and the embedding of the object $\mathbf{e}_o$, using the $L_1$ or $L_2$ norm $\lvert\lvert \cdot \rvert\rvert$. Such scoring function is then used on positive and negative triples $t^+ \in \mathcal{G}, t^- \in \mathcal{N}$ in the loss function. This is usually a pairwise margin-based loss (Bordes et al., 2013), negative log-likelihood, or multi-class log-likelihood (Lacroix et al., 2018). Since the training set usually includes positive statements, we generate synthetic negatives $t^- \in \mathcal{N}$ required for training. We do so by corrupting one side of the triple at a time (i.e. either the subject or the object), following the protocol proposed by (Bordes et al., 2013).

**Calibration.** Given a knowledge graph embedding model identified by its scoring function $f_m$, with $f_m(t) = \hat{p}$, where $\hat{p}$ is the estimated confidence level that a triple $t = (s, p, o)$ is true, we define $f_m$ to be *calibrated* if $\hat{p}$ represents a true probability. For example, if $f_m(\cdot)$ predicts 100 triples all with confidence $\hat{p} = 0.7$, we expect exactly 70 to be actually true. Calibrating a model requires reliable metrics to detect miscalibration, and effective techniques to fix such distortion. Appendix A.1 includes definitions and background on the calibration metrics adopted in the paper.

# 4    CALIBRATING KNOWLEDGE GRAPH EMBEDDING MODELS PREDICTIONS

We propose two scenario-dependent calibration techniques: we first address the case with ground truth negatives $t^- \in \mathcal{N}$. The second deals with the absence of ground truth negatives.

**Calibration with Ground Truth Negatives.** We propose to use off-the-shelf Platt scaling and isotonic regression, techniques proved to be effective in literature. It is worth reiterating that to calibrate a model negative triples $\mathcal{N}$ are required from a held-out dataset (which could be the validation set). Such negatives are usually available in triple classification datasets (FB13, WN11, YAGO39K)

**Calibration with Synthetic Negatives.** Our main contribution is for the case where no ground truth negatives are provided at all, which is in fact the usual scenario for *link prediction* tasks.

We propose to adopt Platt scaling or isotonic regression and to synthetically generate corrupted triples as negatives, while using sample weights to guarantee that the frequencies adhere to the base rate of the population (which is problem-dependent and must be user-specified). It is worth noting that it is not possible to calibrate a model without implicit or explicit base rate. If it is not implicit on the dataset (the ratio of positives to totals), it must be explicitly provided.

We generate synthetic negatives $\mathcal{N}$ following the standard protocol proposed by (Bordes et al., 2013)[1]: for every positive triple $t = (s, p, o)$, we corrupt one side of the triple at a time (i.e. either the subject $s$ or the object $o$) by replacing it with other entities in $\mathcal{E}$. The number of corruptions generated per positive is defined by the user-defined corruption rate $\eta \in \mathbb{N}$. Since the number of negatives $N = |\mathcal{N}|$ can be much greater than the number of positive triples $P = |\mathcal{G}|$, when dealing with calibration with synthetically generated corruptions, we weigh the positive and negative triples to make the calibrated model match the population base rate $\alpha = P/(P + N) \in [0, 1]$, otherwise the base rate would depend on the arbitrary choice of $\eta$.

Given a positive base rate $\alpha$, we propose the following weighting scheme:

$$
\begin{aligned}
\omega_+ &= \eta \quad \text{for positive triples } \mathcal{G} \\
\omega_- &= \frac{1}{\alpha} - 1 \quad \text{for negative triples } \mathcal{N}
\end{aligned}
\tag{1}
$$

where $\omega_+ \in \mathbb{R}$ is the weight associated to the positive triples and $\omega_- \in \mathbb{R}$ to the negatives. The $\omega_+$ weight removes the imbalance determined by having a higher number of corruptions than positive triples in each batch. The $\omega_-$ weight guarantees that the given positive base rate $\alpha$ is respected.

The above can be verified as follows. For the unweighted problem, the positive base rate is simply the ratio of positive examples to the total number of examples:

$$
\alpha = \frac{P}{P + N}
\tag{2}
$$

If we add uniform weights to each class, we have:

$$
\alpha = \frac{\omega_+ P}{\omega_- N + \omega_+ P}
\tag{3}
$$

By defining $\omega_+ = \eta$, i.e. adopting the ratio of negatives to positives (corruption rate), we then have:

$$
\alpha = \frac{P \frac{N}{P}}{N \omega_- + P \frac{N}{P}} = \frac{N}{\omega_- N + N} = \frac{1}{\omega_- + 1}
\tag{4}
$$

Thus, the negative weights is:

$$
\omega_- = \frac{1}{\alpha} - 1
\tag{5}
$$

# 5    RESULTS

We compute the calibration quality of our heuristics, showing that we achieve calibrated predictions even when ground truth negative triples are not available. We then show the impact of calibrated predictions on the task of triple classification.

---

[1]We also experimented with per-batch entities only, without any significant changes to the results. Future work will experiments with additional techniques as proposed by Kotnis & Nastase (2017).

**Datasets.** We run experiments on triple classification datasets that include ground truth negatives (Table 1). We train on the training set, calibrate on the validation set, and evaluate on the test set.

- **WN11** (Socher et al., 2013). A subset of Wordnet (Miller, 1995), it includes a large number of hyponym and hypernym relations thus including hierarchical structures.
- **FB13** (Socher et al., 2013). A subset of Freebase (Bollacker et al., 2008), it includes facts on famous people (place of birth and/or death, profession, nationality, etc).
- **YAGO39K** (Lv et al., 2018). This recently released dataset has been carved out of YAGO3 (Mahdisoltani et al., 2013), and includes a mixture of facts about famous people, events, places, and sports teams.

We also use two standard link prediction benchmark datasets, **WN18RR** (Dettmers et al., 2018) (a subset of Wordnet) and **FB15K-237** (Toutanova et al., 2015) (a subset of Freebase). Their test sets do not include ground truth negatives.

**Implementation Details.** The knowledge graph embedding models are implemented with the AmpliGraph library (Costabello et al., 2019) version 1.1, using TensorFlow 1.13 (Abadi et al., 2016) and Python 3.6 on the backend. All experiments were run under Ubuntu 16.04 on an Intel Xeon Gold 6142, 64 GB, equipped with a Tesla V100 16GB. Code and experiments are available at `https://github.com/Accenture/AmpliGraph`.

**Hyperparameter Tuning.** For each dataset in Table 1a, we train a TransE, DistMult, and a ComplEx knowledge graph embedding model. We rely on typical hyperparameter values: we train the embeddings with dimensionality $k = 100$, Adam optimizer, initial learning rate $\alpha_0 = $ 1e-4, negatives per positive ratio $\eta = 20$, $epochs = 1000$. We train all models on four different loss functions: Self-adversarial (Sun et al., 2019), pairwise (Bordes et al., 2013), NLL, and Multiclass-NLL (Lacroix et al., 2018). Different losses are used in different experiments.

## 5.1 CALIBRATION RESULTS

**Calibration Success.** Table 2 reports Brier scores and log losses for all our calibration methods, grouped by the type of negative triples they deal with (ground truth or synthetic). All calibration methods show better-calibrated results than the uncalibrated case, by a considerable margin and for all datasets. In particular, to put the results of the synthetic strategy in perspective, if we suppose to predict the positive base rate as a baseline, for each of the cases in Table 2 (the three datasets share the same positive base rate $\alpha = 0.5$), we would get Brier score $B = 0.25$ and log loss $L_{log} = 0.69$, results that are always worse than our methods. There is considerable variance of results between models given a dataset, which also happens when varying losses given a particular combination of model and dataset (Table 3). TransE provides the best results for WN11 and FB13, while DistMult works best for YAGO39K. We later propose that this variance comes from the quality of the embeddings themselves, that is, better embeddings allow for better calibration.

In Figure 2, we also evaluate just the frequencies themselves, ignoring sharpness (i.e. whether probabilities are close to 0 or 1), using reliability diagrams for a single model-loss combination, for all datasets (ComplEx+NLL). Calibration plots show a remarkable difference between the uncalibrated baseline (s-shaped blue line on the left-hand side) and all calibrated models (curves closer to the identity function are better). A visual comparison of uncalibrated curves in Figure 1 with those in Figure 2 also gives a sense of the effectiveness of calibration.

**Ground Truth vs Synthetic.** As expected, the ground truth method generally performs better than the synthetic calibration, since it has more data in both quantity (twice as much) and quality (two classes instead of one). Even so, the synthetic method is much closer to the ground truth than to the uncalibrated scores, as highlighted by the calibration plots in Figure 2. For WN11, it is actually as good as the calibration with the ground truth. This shows that our proposed method works as intended and could be used in situations where we do not have access to the ground truth, as is the case for most knowledge graph datasets.

**Isotonic vs Platt.** Isotonic regression performs better than Platt scaling in general, but in practice Isotonic regression has the disadvantage of not being a convex or differentiable algorithm Zadrozny & Elkan (2002). This is particularly problematic for the synthetic calibration, as it requires the generation of the synthetic corruptions, which can only be made to scale via a mini-batch based

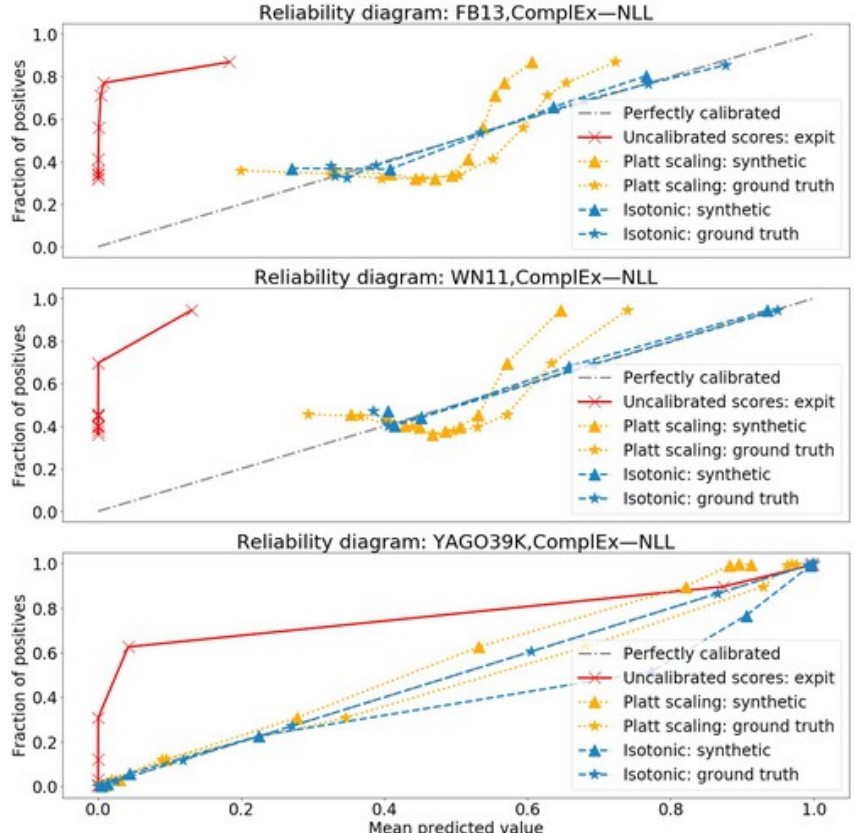

Figure 2: Calibration plots for the best calibrated model-loss combinations. Isotonic regression delivers the best results, getting very close to the perfectly calibrated line, both when used with the ground truth method or our proposed synthetic method. Best viewed in colors.

| | | **Brier Score** | | | | **Log Loss** | | | | |
|---|---|---|---|---|---|---|---|---|---|---|
| | | | Ground Truth | | Synthetic | | | Ground Truth | | Synthetic | |
| | | Uncalib | Platt | Iso | Platt | Iso | Uncalib | Platt | Iso | Platt | Iso |
| **WN11** | TransE | .443 | .089 | **.087** | .092 | .088 | 1.959 | .302 | **.295** | .311 | .296 |
| | DistMult | .488 | .213 | .208 | .214 | .208 | 5.625 | .618 | .604 | .618 | .601 |
| | ComplEx | .490 | .240 | .227 | .240 | .228 | 6.061 | .674 | .651 | .674 | .650 |
| | HolE | .474 | .235 | .235 | .235 | .236 | 2.731 | .663 | .661 | .663 | .668 |
| **FB13** | TransE | .446 | **.124** | **.124** | .148 | .141 | 1.534 | **.390** | .391 | .459 | .442 |
| | DistMult | .473 | .178 | .170 | .185 | .192 | 2.177 | .533 | .518 | .549 | .567 |
| | ComplEx | .481 | .177 | .170 | .182 | .189 | 2.393 | .534 | .516 | .544 | .565 |
| | HolE | .452 | .229 | .228 | .242 | .263 | 1.681 | .650 | .651 | .677 | .725 |
| **YAGO 39K** | TransE | .363 | .095 | .093 | .106 | .110 | 1.062 | .319 | .309 | .370 | .376 |
| | DistMult | .284 | .081 | **.079** | .093 | .089 | 1.043 | .279 | **.266** | .311 | .308 |
| | ComplEx | .264 | .089 | .084 | .097 | .095 | 1.199 | .305 | .278 | .323 | .313 |
| | HolE | .345 | .141 | .140 | .166 | .162 | 1.065 | .444 | .438 | .581 | .537 |

Table 2: Calibration test results (self-adversarial loss (Sun et al., 2019)). Low score = better. Best results in bold for each combination of dataset and metric.

optimization procedure. Platt scaling, given that it is a convex and differentiable loss, can be made part of a computational graph and optimized with mini-batches, thus it can rely on the modern computational infrastructure designed to train deep neural networks.

**Influence of Loss Function.** We experiment with different losses, to assess how calibration affects each of them (Table 3). We choose to work with TransE, which is reported as a strong baseline in (Hamaguchi et al., 2017). Self-adversarial loss obtains the best calibration results for all calibra-

| $\mathcal{L}$ | Brier Score | | | | Log Loss | | | | MRR (filtered) |
|---|---|---|---|---|---|---|---|---|---|
| | Ground Truth | | Synthetic | | Ground Truth | | Synthetic | | |
| | Platt | Iso | Platt | Iso | Platt | Iso | Platt | Iso | |
| Pairwise | .202 | .198 | .209 | .200 | .591 | .585 | .606 | .589 | .058 |
| NLL | .093 | .088 | .094 | .088 | .342 | .299 | .344 | .301 | .134 |
| Multiclass-NLL | .204 | .189 | .204 | .189 | .599 | .550 | .599 | .551 | .108 |
| Self-adversarial | .089 | **.087** | .092 | .088 | .302 | **.295** | .311 | .296 | **.155** |

(a) WN11

| $\mathcal{L}$ | Brier Score | | | | Log Loss | | | | MRR (filtered) |
|---|---|---|---|---|---|---|---|---|---|
| | Ground Truth | | Synthetic | | Ground Truth | | Synthetic | | |
| | Platt | Iso | Platt | Iso | Platt | Iso | Platt | Iso | |
| Pairwise | .225 | .203 | .225 | .208 | .636 | .582 | .637 | .594 | .282 |
| NLL | .209 | .203 | .240 | .244 | .614 | .592 | .676 | .685 | .202 |
| Multiclass-NLL | .146 | .146 | .162 | .159 | .455 | .454 | .500 | .490 | **.402** |
| Self-adversarial | **.124** | **.124** | .142 | .141 | **.390** | **.390** | .446 | .442 | .296 |

(b) FB13

| $\mathcal{L}$ | Brier Score | | | | Log Loss | | | | MRR (filtered) |
|---|---|---|---|---|---|---|---|---|---|
| | Ground Truth | | Synthetic | | Ground Truth | | Synthetic | | |
| | Platt | Iso | Platt | Iso | Platt | Iso | Platt | Iso | |
| Pairwise | .123 | .103 | .147 | .113 | .445 | .352 | .477 | .393 | **.371** |
| NLL | .187 | .170 | .260 | .200 | .577 | .518 | .756 | .622 | .063 |
| Multiclass-NLL | .111 | .104 | .128 | .116 | .392 | .350 | .431 | .440 | .325 |
| Self-adversarial | .095 | **.093** | .113 | .109 | .319 | **.308** | .399 | .376 | .169 |

(c) YAGO39K

Table 3: Calibration test results using different losses $\mathcal{L}$ with TransE. Lower calibration metrics = better. We compute MRR only on positive test triples. Self-adversarial loss achieves better calibrated results across the board. Results show no correlation between MRR and calibration performance, i.e. embeddings that bring higher MRR are not necessarily easier to calibrate. Best results in bold.

tion methods, across all datasets. Experiments also show the choice of the loss has a big impact, greater than the choice of calibration method or embedding model. We assess whether such variability is determined by the quality of the embeddings. To verify whether better embeddings lead to sharper calibration, we report the mean reciprocal rank (MRR), which, for each true test triple, computes the (inverse) rank of the triple against synthetic corruptions, then averages the inverse rank (Table 3). In fact, we notice no correlation between calibration results and MRR. In other words, embeddings that lead to the best predictive power are not necessary the best calibrated.

**Positive Base Rate.** We apply our synthetic calibration method to two link prediction benchmark datasets, FB15K-237 and WN18RR. As they only provide positive examples, we apply our method with varying base rates $\alpha_i$, linearly spaced from $0.05$ to $0.95$. We evaluate results relying on the closed-world assumption, i.e. triples not present in training, validation or test sets are considered negative. For each $\alpha_i$ we calibrate the model using the synthetic method with both isotonic regression and Platt scaling. We sample negatives from the negative set under the implied negative rate, and calculate a baseline which is simply having all probability predictions equal to $\alpha_i$. Figure 3 shows that isotonic regression and Platt scaling perform similarly and always considerably below the baseline. As expected from the previous results, the uncalibrated scores perform poorly, only reaching acceptable levels around some particular base rates.

**Triple Classification and Decision Threshold.** To overcome the need to learn $|\mathcal{R}|$ decision thresholds $\tau_i$ from the validation set, we propose to rely on calibrated probabilities, and use the natural threshold of $\tau = 0.5$. Table 4 shows how calibration affects the triple classification task, comparing with the literature standard of per-relation thresholds (last column). For simplicity, note we use the same self-adversarial loss in Table 2 and Table 4. We learn thresholds $\tau_i$ on validation sets, resulting in 11, 7, and 33 thresholds for WN11, FB13 and YAGO39K respectively.

Using a single $\tau = 0.5$ and calibration provides competitive results compared to multiple learned thresholds (note uncalibrated results with $\tau = 0.5$ are poor, as expected). It is worth mentioning that

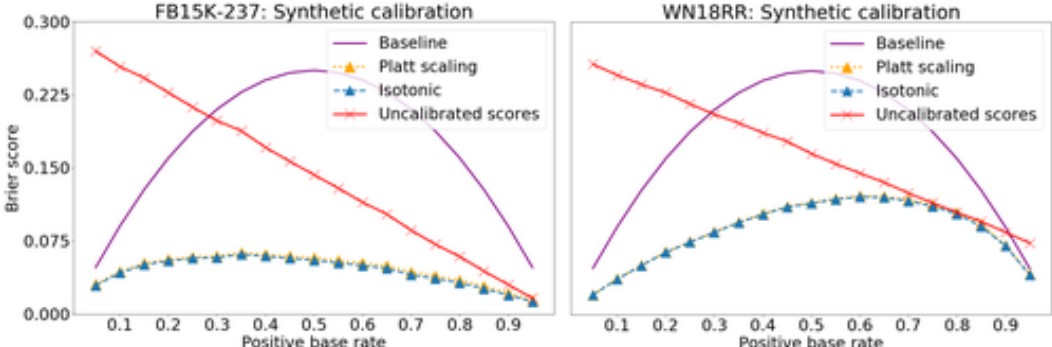

Figure 3: Synthetic calibration on FB15K-237 and WN18RR, with varying positive base rates. The baseline stands for using the positive base rate as the probability prediction. Results are evaluated under the closed-world assumption, using the same positive base rate used to calibrate the models.

| | | Ground Truth ($\tau = .5$) | | Synthetic ($\tau = .5$) | | Uncalib. | Uncalib. (Per-Relation $\tau$) | |
|---|---|---|---|---|---|---|---|---|
| | | Platt | Iso | Platt | Iso | ($\tau = .5$) | Reproduced | Literature |
| | TransE | 88.8 | **88.9** | **88.9** | 88.9 | 50.7 | 88.2 | |
| | DistMult | 66.5 | 67.2 | 66.4 | 67.1 | 50.8 | 67.2 | **88.9**[*] |
| **WN11** | ComplEx | 60.6 | 62.4 | 60.0 | 62.4 | 50.8 | 59.6 | |
| | HolE | 59.3 | 59.0 | 59.3 | 59.0 | 50.9 | 60.8 | |
| | TransE | 82.4 | 82.4 | 80.7 | 80.2 | 50.0 | 82.1 | |
| | DistMult | 72.5 | 73.2 | 72.1 | 70.2 | 50.1 | 80.8 | **89.1**[*] |
| **FB13** | ComplEx | 73.8 | 74.2 | 74.2 | 72.4 | 50.1 | 83.6 | |
| | HolE | 60.3 | 60.6 | 57.8 | 54.3 | 50.0 | 62.6 | |
| | TransE | 87.2 | 87.8 | 85.3 | 84.9 | 50.2 | 88.8 | |
| **YAGO 39K** | DistMult | 88.9 | 89.3 | 88.1 | 88.5 | 56.7 | 90.2 | **93.8**[†] |
| | ComplEx | 87.3 | 88.2 | 86.9 | 87.2 | 61.1 | 89.4 | |
| | HolE | 80.4 | 80.4 | 78.4 | 78.5 | 50.6 | 81.5 | |

Table 4: Effect of calibration on triple classification accuracy. Best results in bold. For all calibration methods there is one single threshold, $\tau = 0.5$. For the per-relation $\tau$, we learned multiple thresholds from validation sets (Appendix A.5). We did not carry out additional model selection, and used Table 2 hyperparameters instead. Isotonic regression reaches state-of-the-art results for WN11. Results of * from (Zhang et al., 2018); ⋆ from (Ji et al., 2016); † from (Lv et al., 2018).

we are at par with state-of-the-art results for WN11. Isotonic regression is again the best method, but there is more variance in the model choice. Our proposed calibration method with synthetic negatives performs well overall, even though calibration is performed only using half of the validation set (negatives examples are replaced by synthetic negatives).

## 6 CONCLUSION

We propose a method to calibrate knowledge graph embedding models. We target datasets with and without ground truth negatives. We experiment on triple classification datasets and apply Platt scaling and isotonic regression with and without synthetic negatives controlled by our heuristics. All calibration methods perform significantly better than uncalibrated scores. We show that isotonic regression brings better calibration performance, but it is computationally more expensive. Additional experiments on triple classification shows that calibration allows to use a single decision threshold, reaching state-of-the-art results without the need to learn per-relation thresholds.

Future work will evaluate additional calibration algorithms, such as beta calibration (Kull et al., 2017) or Bayesian binning (Naeini et al., 2015). We will also experiment on ensembling of knowledge graph embedding models, inspired by(Krompaß & Tresp, 2015). The rationale is that different models operate on different scales, but calibrating brings them all to the same probability scale, so their output can be easily combined.

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

# A    APPENDIX

## A.1    CALIBRATION METRICS

**Reliability Diagram** (DeGroot & Fienberg, 1983; Niculescu-Mizil & Caruana, 2005). Also known as *calibration plot*, this diagram is a visual depiction of the calibration of a model (see Figure 1 for an example). It shows the expected sample accuracy as a function of the estimated confidence. A hypothetical perfectly calibrated model is represented by the diagonal line (i.e. the identity function). Divergence from such diagonal indicates calibration issues (Guo et al., 2017).

**Brier Score** (Brier, 1950). It is a popular metric used to measure how well a binary classifier is calibrated. It is defined as the mean squared error between $n$ probability estimates $\hat{p}$ and the corresponding actual outcomes $y \in 0, 1$. The smaller the Brier score, the better calibrated is the model. Note that the Brier score $B \in [0, 1]$.

$$B = \frac{1}{n} \sum_{i=1}^{n} (y_i - \hat{p}_i)^2 \tag{6}$$

**Log Loss** is another effective and popular metric to measure the reliability of the probabilities returned by a classifier. The logarithmic loss measures the relative uncertainty between the probability estimates produced by the model and the corresponding true labels.

$$L_{log} = -(y \cdot log(\hat{p}) + (1 - y) \cdot log(1 - \hat{p})) \tag{7}$$

**Platt Scaling**. Proposed by (Platt et al., 1999) for support vector machines, Platt scaling is a popular parametric calibration techniques for binary classifiers. The method consists in fitting a logistic regression model to the scores returned by a binary classifier, such that $\hat{q} = \sigma(a\hat{p} + b)$, where $\hat{p} \in \mathbb{R}$ is the uncalibrated score of the classifier, $a, b \in \mathbb{R}$ are trained scalar weights. and $\hat{q}$ is the calibrated probability returned as output. Such model can be trained be trained by optimizing the NLL loss with non-binary targets derived by the Bayes rule under an uninformative prior, resulting in an Maximum a Posteriori estimate.

**Isotonic Regression** (Zadrozny & Elkan, 2002). This popular non-parametric calibration techniques consists in fitting a non-decreasing piecewise constant function to the output of an uncalibrated classifier. As for Platt scaling, the goal is learning a function $\hat{q} = g(\hat{p})$, such that $\hat{q}$ is a calibrated probability. Isotonic regression learns $g$ by minimizing the square loss $\sum_{i=1}^{n} (\hat{q}_i - y_i)^2$ under the constraint that $g$ must be piecewise constant (Guo et al., 2017).

## A.2    CALIBRATION DIAGRAMS: INSTANCES PER BIN

We present in Figure 4 the total count of instances for each bin used in the calibration plots included in Figure 2. As expected, calibration considerably helps spreading out instances across bins, whereas in uncalibrated scenarios instances are squeezed in the first or last bins.

## A.3    IMPACT OF MODEL HYPERPARAMETERS: $\eta$ AND EMBEDDING DIMENSIONALITY

In Figure 5 we report the impact of negative/positive ratio $\eta$ and the embedding dimensionality $k$. Results show that the embedding size $k$ has higher impact than the negative/positive ratio $\eta$. We observe that calibrated and uncalibrated low-dimensional embeddings have worse Brier score. Results also show that any $k > 50$ does not improve calibration anymore. The negative/positive ratio $\eta$ follows a similar pattern: choosing $\eta > 10$ does not have any effect on the calibration score.

## A.4    POSITIVE BASE RATE EXPERIMENTS: LINK PREDICTION PERFORMANCE

In Table 5, we present the traditional knowledge graph embedding rank metrics: MRR (mean reciprocal rank), MR (mean rank) and Hits@10 (precision at the top-10 results). We report the results for all datasets and models used in the main text, which appear in Table 2, Table 4 and Figure 3.

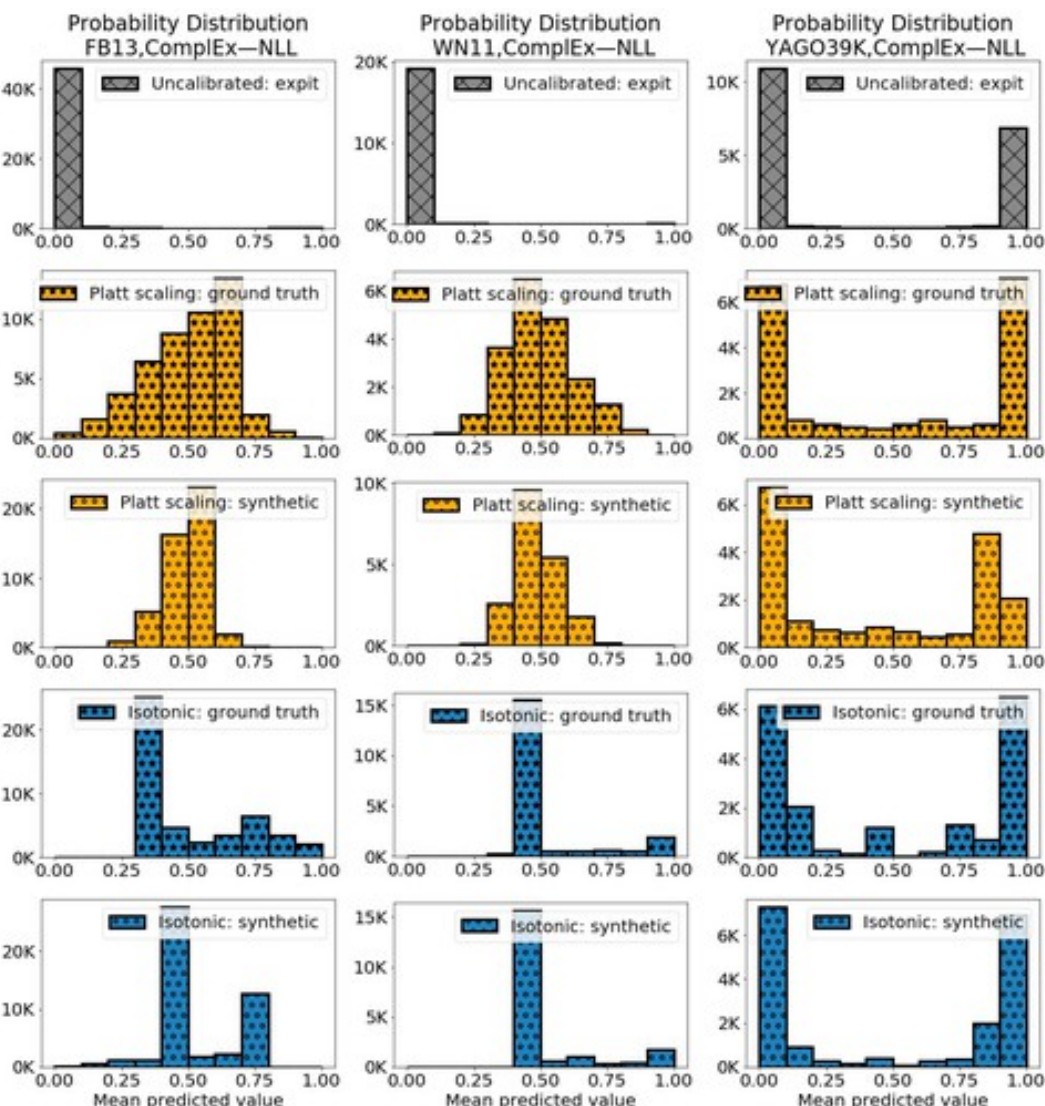

Figure 4: Histograms show the total count instances for each bin used by calibration plots presented in Figure 2. Best viewed in colors.

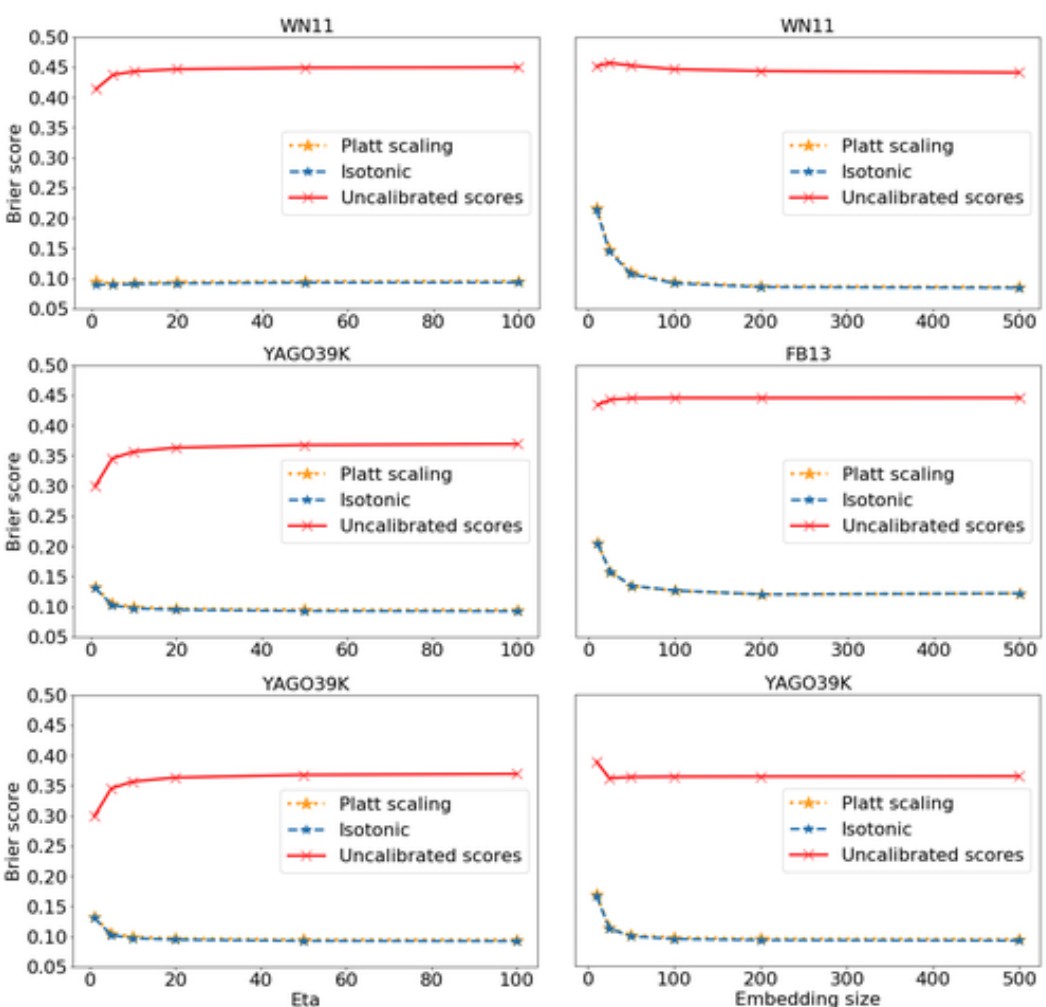

Figure 5: Impact of $\eta$ (eta) and $k$ (embedding size) on the Brier score. We used TransE and the Self-Adversarial loss for all datasets. Best viewed in colors.

|  |  | MR | MRR | Hits@10 |
|---|---|---|---|---|
| **WN11** | TransE | 2289 | .155 | .309 |
|  | DistMult | 10000 | .045 | .081 |
|  | ComplEx | 13815 | .054 | .094 |
|  | HolE | 13355 | .017 | .035 |
| **FB13** | TransE | 3431 | .296 | .394 |
|  | DistMult | 6667 | .183 | .337 |
|  | ComplEx | 8937 | .018 | .039 |
|  | HolE | 8937 | .018 | .039 |
| **YAGO39K** | TransE | 244 | .169 | .319 |
|  | DistMult | 635 | .306 | .620 |
|  | ComplEx | 1074 | .531 | .753 |
|  | HolE | 922 | .101 | .189 |
| **WN18RR** | ComplEx | 4111 | .506 | .583 |
| **FB15K-237** | ComplEx | 183 | .320 | .499 |

Table 5: Standard *filtered* metrics for knowledge graph embeddings models. The models are implemented in the same codebase and share the same evaluation protocol. Note that we do not include results from *reciprocal* evaluation protocols.

## A.5 PER-RELATION DECISION THRESHOLDS

We report in Table 6 the per-relation decision thresholds $\tau$ used in Table 4, under the 'Reproduced' column. Note that the thresholds reported here are not probabilities, as they have been applied to the raw scores returned by the model-dependent scoring function $f_m(t)$.

| Relation | $\tau$ |
|---|---|
| _domain_region | -6.0069733 |
| _domain_topic | -5.5207396 |
| _has_instance | -6.2901406 |
| _has_part | -5.673306 |
| _member_holonym | -6.3117476 |
| _member_meronym | -5.982978 |
| _part_of | -5.798244 |
| _similar_to | -6.852225 |
| _subordinate_instance_of | -5.4750223 |
| _synset_domain_topic | -6.6392403 |
| _type_of | -6.743014 |

WN11

| Relation | $\tau$ |
|---|---|
| cause_of_death | -3.5680597 |
| ethnicity | -3.4997067 |
| gender | -3.4051323 |
| institution | -3.547462 |
| nationality | -3.8507419 |
| profession | -3.7040129 |
| religion | -3.5918012 |

FB13

| Relation | $\tau$ | Relation | $\tau$ |
|---|---|---|---|
| 0 | -3.9869666 | 16 | -1.8443029 |
| 1 | -3.6161883 | 17 | -3.4323683 |
| 2 | -2.9660778 | 18 | -1.6325312 |
| 3 | -2.9241138 | 19 | -4.2211304 |
| 4 | -3.8640308 | 20 | -4.101904 |
| 5 | -3.685308 | 21 | -3.840962 |
| 6 | -2.861393 | 22 | -1.832546 |
| 7 | -3.3280334 | 23 | -2.0101485 |
| 8 | -3.0741293 | 24 | -3.1512089 |
| 9 | -3.1950998 | 25 | -2.4524217 |
| 10 | -2.951118 | 27 | -3.4848583 |
| 11 | -1.8720441 | 29 | -2.4269128 |
| 12 | -2.4230814 | 31 | -2.209188 |
| 13 | -1.542841 | 32 | -1.3310984 |
| 14 | -2.6944544 | 33 | -2.3231838 |
| 15 | -3.381497 | 35 | -2.0017974 |
| 36 | -1.3954651 | | |

YAGO39K

Table 6: Relation-specific decision thresholds learned on uncalibrated raw scores (See also Table 4 for a report on triple classification results.)

