# OpenReview forum: "Probability Calibration for Knowledge Graph Embedding Models"
_ICLR.cc/2020/Conference — Accept (Poster)_

### Official Review · AnonReviewer2 · 2019-10-22
**Official Blind Review #2**

**Rating:** 6

**Review:**

In this paper, the authors deal with the calibration problem in graph embedding models. They used Platt scaling and isotonic regression in the situation when there is ground truth negatives. They also work when there is no ground truth negatives. In this situation, they proposed a calibration heuristics for synthetically generated negatives. Overall, the approach is not very innovative, but the problem they tackled is under studied. The presentation of the whole paper is ok, although it falls onto preliminary side. Since I did not identify any technical problem so far, I will vote for a weak acceptance, unless I observe more technical issues during the discussion.



**Experience Assessment:**

I have read many papers in this area.

**Review Assessment: Checking Correctness Of Derivations And Theory:**

I assessed the sensibility of the derivations and theory.

**Review Assessment: Checking Correctness Of Experiments:**

I assessed the sensibility of the experiments.

**Review Assessment: Thoroughness In Paper Reading:**

I read the paper at least twice and used my best judgement in assessing the paper.

---

> ### Author Response · Authors · 2019-11-15
> **Thanks for your comments**
>
> Thank you for the review. We have improved the preliminary section as suggested.

---

### Official Review · AnonReviewer4 · 2019-11-01
**Official Blind Review #4**

**Rating:** 3

**Review:**

This paper focuses on the calibration of the knowledge graph embedding task with Platt scaling and isotonic regression. This paper is well-written, well-motivated and well-organized. However, my major concern is the novelty of this paper or the contribution.

Major Concerns:

1. This paper lacks novelty. In this paper, the authors only apply the existing techniques (e.g. Platt Scaling, Isotonic Regression) to tackle the calibration issue, which makes a minor contribution. I suggest that the authors could provide their own method specified to knowledge graph tasks rather than leverage the off-shelf methods.

2. The related work could be enhanced, while the preliminaries could be reduced. Actually, in the area of knowledge graph or natural language processing, the preliminary of this paper is a bit trivial.

Minor Concerns:

1. In Table 2, we can conclude that Iso will be better than Platt in general. However, in the case of FB13 (ComplEx) and YAGO (TransE), the results are again the conclusion. Is this because of the optimization issue? I suggest the authors clearly state the experimental analysis.

**Experience Assessment:**

I have published in this field for several years.

**Review Assessment: Checking Correctness Of Derivations And Theory:**

I carefully checked the derivations and theory.

**Review Assessment: Checking Correctness Of Experiments:**

I assessed the sensibility of the experiments.

**Review Assessment: Thoroughness In Paper Reading:**

I read the paper thoroughly.

---

> ### Author Response · Authors · 2019-11-15
> **Our paper novelty and other comments**
>
> Major:
>
> 1 [Novelty]. The paper is novel because it proposes the first framework for calibrating knowledge graph embedding models, something that to the best of our knowledge has never been done before, besides en passant comments in one workshop paper (Krompaß and Tresp, 2015). The proposal of calibrating without positives is certainly a novel contribution as well, plus the use of sample weights to correct the distributional distortions created by the corruption generation procedure.
>
> 2. [Shorten preliminaries]: We shortened the preliminaries as suggested, and moved calibration-related background to appendix A.1. Our main goal is first of all raise awareness on the problem of calibration in the graph representation learning community, and we believe some preliminaries are useful for better engagement with the reader, given that our community has not addressed the issue until now.
>
> 3. [insufficient literature review] As pointed out at the beginning of Section 2, a comprehensive survey is out of the scope of this work. Nevertheless, we added many extra references in Section 2.
>
>
> Minor:
>
> [Table2, analysis of results]: yes, experiments show that in general Isotonic regression is better than Platt scaling. We are not sure to understand what reviewer #4 means by "the results are again the conclusion. Is this because of the optimization issue?". The message we want to send in Table2 is that our calibration works across  different datasets and models. We always obtain better Brier score and log loss. We also show that our heuristic based on synthetic negatives always obtains better calibration scores, regardless of the dataset and model adopted. We hope to have answered your doubts.

---

### Official Review · AnonReviewer5 · 2019-11-02
**Official Blind Review #5**

**Rating:** 8

**Review:**


1. Summary
The paper studies probability calibration for three different knowledge graph embedding methods, with a focus on TransE evaluated on the task of knowledge graph triple classification. It studies Brier and log loss performance of Platt scaling and isotonic regression probability calibration on WN11 for TransE and claims that better calibration yields better performance as measured by mean reciprocal rank. Calibration plots for other datasets are also included as evidence. Furthermore, evidence is presented that probability calibration can lead to better performance for the task of triple classification. The main contributions of the paper also include the adaption of sampling techniques introduced by Bordes et al. (2013) adapted for estimating negatives for probability calibrations.

2. Decision (See the updated decision in the comment below)

Probability calibration is a very relevant issue, particularly in industry and when combining knowledge graph embedding models as external data in other models. Thus I see this work as a valuable contribution to the literature. In particular, I like the analysis from multiple views: Calibration plots, calibration metrics, and model performance. However, there is currently not enough evidence in the paper to make recommendations or judgments about when researchers and practitioners may want to use probability calibration. I also believe the datasets and models are not well tied into the literature, for example, in 2018/2019 I can find 3 papers for triple classification and 9 papers for link prediction as triple/entity ranking and from the data, it is not clear how probability calibration affects the latter. In the current state of the work, I recommend rejecting this work.

3. Further supporting arguments

As a researcher and practitioner in this area, I know very well that predictions of most knowledge graph embedding models usually live near the decision boundary so that there is little difference between probability or score of a true positive and false positive. Also talking with people in industry, I heard that word embedding models are currently not that useful practically because they make too many useless predictions. This shows me that probability calibration is an important topic and I see this study as an important contribution to the field which is often mindlessly following evaluation metrics.

However, from experience I also know that evaluation of knowledge graph embedding methods is not very reliable, that is people often get widely varying results and replication is difficult. Thus it is difficult to trust results if they are not well tied into the literature and compared against multiple datasets and models. This work focuses on three models TranseE, DistMult and ComplEx. DistMult and ComplEx have become models that are viewed as quite reliable to compare against. However, their performance is mostly studied on a learning-to-rank objective on datasets such as FB15k-237 and WN18RR. The authors report Brier score for their synthetic calibration method these datasets, but do not report any modeling results. Inclusions of results on these datasets would greatly improve this work.

The authors also currently focus on establishing that probability calibration improves the performance of the models. They claim that low Brier score or log loss are tied to good performance, but Pairwise and Multiclass-NLL loses achieve similar Brier/log loss performance while the MRR is double for Multiclass-NLL compared to the Pairwise loss. NLL and Multiclass NLL losses have similar MRR but very different Brier/log loss performance. As such I do not think this claim is sufficiently substantiated. I do not believe it is necessary to establish that better probability calibration is correlated with better model performance. I view the careful study of probability calibration and its effects per se as more useful.
As mentioned above, I also believe the results on WN11, FB13, and YAGO39k to not be sufficient to evaluate the effect of probability calibration.

4. Additional feedback

I really like this work. I think adding more results would make this paper great and I would be happy to change my acceptance decision.

As mentioned above I believe including results on FB15k-237 and WN18RR would make the results easier to interpret. Please also add more results to the table (no need to rerun those experiments, take them from other papers). I really like the analysis of Brier Score/Log loss and MRR. I think if you would extend this it would give very valuable insights into how probability calibration relates to performance.

One additional experiment which I do not deem critical, but which would improve your work further would be to tie probability calibration into a more practical setting. A setting that is also very interesting to researchers is if probability calibration would affect the results in tasks where you use word embedding models as an external "knowledge source". I really like  Kumar et al., 2019[1] since their word embedding model integrated into an entity linking model beats a strong BERT baseline. But I think a study of any task/model of your choice that integrates a knowledge embedding model would be a valuable addition to your work.

Again as I mentioned above, I do not believe it is critical to show improve performance on these tasks, a study of the effects of probability calibration is valuable in its own right. You might want to slightly pivot into this direction if you have sufficient evidence to make judgments about the effects of probability calibration.

Further small details: In the introduction, you make specific claims and justify them by citing a survey paper (Nickel et al., 2016). It would be easier for the reader to look up these claims in the source rather than in the survey paper. I believe there is a typo in your derivation in equation (6): the denominator of the second term should be just w-N + N or N(w- + 1).

[1] Zero-shot Word Sense Disambiguation using Sense Definition Embeddings: https://www.aclweb.org/anthology/P19-1568/

**Experience Assessment:**

I have published one or two papers in this area.

**Review Assessment: Checking Correctness Of Derivations And Theory:**

I assessed the sensibility of the derivations and theory.

**Review Assessment: Checking Correctness Of Experiments:**

I carefully checked the experiments.

**Review Assessment: Thoroughness In Paper Reading:**

I read the paper at least twice and used my best judgement in assessing the paper.

---

> ### Author Response · Authors · 2019-11-15
> **Our reply to your suggestions**
>
>
> - [Unsubstantiated claim: impact of calibration on link prediction] There is a small misunderstanding: we did not claim there is a causal effect between calibration and rank metrics such as MR, MRR or Hits@K (in fact, calibration does not change the rank-order of the link prediction results). Our initial submission limited to hint at a possible correlation between calibration scores and MRR (Table3 caption).
> In fact, as suggested by reviewer #4, we carried out additional experiments which are included in the latest revision attached, and our updated results in Table 3 suggest that there is no correlation between calibration results and MRR, i.e. "better" embeddings (i.e. embeddings that lead to higher link prediction MRR) are not necessarily easier to calibrate.
> We have clarified this in the main text (Sect 5.1 and Table3 caption).
>
> On the other hand, calibration does affect triple classification. More precisely, it affects the way we choose the decision threshold \tau. We show that with calibrated probabilities you only need one natural decision threshold \tau=0.5 to maximize accuracy, while other methods require arbitrary per-relation thresholds.
>
>
> - [no evidence of real-world use of calibration] As pointed out in the introduction, the usefulness of calibration lies on being able to trust the output of knowledge graph embedding models and even quantify this trust. This has great importance when discovery new links in biological networks: better calibrated probabilities help human experts (biologists) validating discoveries and make a ML pipeline based on graph embeddings more trustworthy.
> Moreover, a minor application is triple classification, where calibrated probabilities replace per-relation thresholds (Section 5.1, Table 4).
> We improved the part in the introduction where we talk about real-world examples of why calibration is important.
>
> - [insufficient literature review] As pointed out at the beginning of Section 2, a comprehensive survey is out of the scope of this work. However, we have added extra references and enriched the prior art section.
>
> - [Main claim of the paper is calibration study]: We would like to point out again that this is the only goal of our work. We do not aim at showing causality between calibration and predictive power - as stated above. We have made clearer this point in the text (Sect 5.1).
>
> - [Add link prediction metrics results] In the appendix, we added a table with MRR, MR, Hits@10 for all the datasets used. As pointed out in A.1) above, we do not claim any causal relation of calibration on such task metrics.
>
> Minor comments:
>
> [Typo in Equation 6] Fixed, thanks.
>
> [Reference to Nickel et al. 2016 in introduction] We only refer to this work to point the reader to the first paper to suggest the use of a sigmoid function to turn scores into probabilities.
>
> [Additional experiments, knowledge injection]: in fact, calibration does not affect the embeddings per-se, as it consists in a downstream operation carried out after training. If Platt scaling is adopted, then new weights are learned, but these are separate from the embeddings, which are not touched at this stage. That means calibration will not have any impact on such experiment.

---

### Official Review · AnonReviewer6 · 2019-11-04
**Official Blind Review #6**

**Rating:** 6

**Review:**

This is the first work that studies probability calibration for knowledge graph embedding models. In the case where ground-truth negatives are available the authors directly use off-the-shelf established calibration techniques (Platt scaling, isotonic regression). When ground-truth negatives are not available they propose to synthetically generate corrupted triples as negatives and use sample weights to guarantee that the frequencies adhere to the base rate.

In general the paper is well-written and easy to follow. Given that the paper's major contribution is experimental insight, and there are no major technical contributions, I would have liked to see a more in-depth analysis of how some of the key hyper-parameters influence the calibration of a model beyond the type of the loss, and beyond the correlation with embedding quality. Overall, I would be willing to increase the score if the authors perform a more comprehensive experimental analysis.

Suggestions to improve the paper:
1) I would expect that especially the negatives per positive ratio \eta, and the dimensionality of the embeddings have a significant impact on model calibration. It would be valuable to experimentally quantify the impact of these key hyper-parameters.
2) It is currently difficult to judge how well-calibrated are the models from the reliability diagrams/calibration plots since the total counts are not shown (e.g. total number of instances with mean predicted value between 0.4 and 0.5). That is, it could be that deviation from identity is due to small sample effects, i.e. we are estimating the fraction of positives from a handful of instances. Showing the total counts for each bin will help the reader better understand the calibration of the models.
3) Several questions can be clarified regarding the sample weights:
3.1) How essential is the proposed weighting scheme? How do the calibration techniques perform when using synthetic negatives with uniform sample weights?
3.2) How does the proposed weighting scheme relate to the the general problem of calibrating models that have class imbalance?
4.1) Can we observe significant difference in terms of calibration between translational distance models and semantic matching models, i.e. using distance-based scoring functions vs. using similarity-based scoring functions. If so is there any reason for that? To help answer this question the authors could compare additional models from each group (beyond the three models used in the paper).
4.2) Are methods that represent entities as random variable to capture uncertainties (e.g. KG2E) better calibrated?
5) Platt scaling assumes that per-class probabilities are normally distributed, while isotonic regression makes no assumption about the input probabilities. Given that Platt scaling performs worse in the experiments it would be interesting to investigate whether this can be (partly) explained by a deviation from the above assumption.
6) Results reported in Table 3 are for WN11. It would be valuable to report similar results for the other datasets in the appendix.
7) it would be beneficial to explore the different procedures proposed in the literature for generating synthetic negatives and their impact on the calibration.

Suggestions to improve the paper that did not impact the score:
1) On the triple classification task in Table 4, there is a significant gap between the literature results and the reproduced results on FB13 and YAGO39K. Is there an explanation for this? Furthermore, it would be interesting to investigate how much do the per-relation \tau_i's deviate from 0.5 when they are learned using both non-calibrated and calibrated probabilities.
2) In Eq. 6 after the second equality shouldn't there be "N/(w- + N)" instead of "N/(w_{-} + PN)"?  Is the additional P a typo?
3) It would be nice to make the figures more readable (e.g. when printed in black and white) by using different markers for each line.

Edit: Rating updated to 6 after rebuttal.

**Experience Assessment:**

I have read many papers in this area.

**Review Assessment: Checking Correctness Of Derivations And Theory:**

I carefully checked the derivations and theory.

**Review Assessment: Checking Correctness Of Experiments:**

I carefully checked the experiments.

**Review Assessment: Thoroughness In Paper Reading:**

I read the paper thoroughly.

---

> ### Author Response · Authors · 2019-11-15
> **Our reply to your suggestions**
>
> Main points:
>
> 1. We ran experiments to assess the impact of the embedding size and negatives/positive ratio \eta. We added such additional results to appendix A.3. Results show that the embedding size has higher impact than the negative/positive ratio \eta. We observe that calibrated and uncalibrated low-dimensional embeddings have worse Brier score. Results also show that any k>50 does not improve calibration anymore. The negative/positive ratio \eta follows a similar pattern: choosing \eta>10 does not have any effect on the calibration score.
>
> 2. In appendix A.2 We added histograms that show the total count of instances for each bin used in the calibration plots. As expected, calibration considerably helps spreading out instances across bins, whereas in uncalibrated scenarios instances are squeezed in the first or last bins.
>
> 3.1. Without sample weights, the base rate will be determined implicitly by the negatives/positive ratio eta used for calibration. For example, if we use eta=3 for calibration, this implies a positive base rate \alpha=25%. As this base rate will most likely be wrong, calibration without sample weights leads to meaningless results.
>
> 3.2. Sample weights allow the user to balance the positives and negatives in a way he or she sees fit for the problem, independent from choices such as the calibration eta (when using corruptions for the calibration). This is indeed similar to dealing with imbalanced datasets, especially in the case where the training dataset distribution does not match the expected test / deployment distribution.
>
> 4.1. We added extra experiments with HolE, another model implemented in the library we used for the experiments. We can certainly add experiments on other models, but as such implementations do not belong to the same codebase, we fear we will most likely end up with unfair comparisons (as you know this is a well-known problem in this community). All in all, the set of models we used is quite diverse (translation-based, tensor-decomposition-based, with different scoring functions), and it is well representative of models actually used in the wild by practitioners, even outside the boundaries of our community.
>
> 4.2. That is an interesting direction for future work. While KG2E proposes to use Gaussian distributed embeddings to account for the uncertainty, their model does not provide the probability of a triple being true, so KG2E would also benefit from the output calibration procedure we propose here. It is an open question how to design embedding methods that naturally lead to well-calibrated probabilities.
>
> 5. Platt scaling was developed originally to calibrate SVMs (Platt et al., 1999), where the output is not a probability, but a continuous score, which is similar to the output scores in knowledge graph embeddings. We could not find a reference that asserts the need of normally distributed class probabilities. Perhaps it was meant that the logits need to be normally distributed, given the connection between Platt scaling and logistic regression. If so, that indeed points to a new direction to investigate the limits of our proposed framework.
>
> 6. We have added two extra tables in 5.1 with additional results for FB13 and YAGO39k.
>
> 7. We have tried two variations of corruption generation, all entities and per-batch entities, without any significant changes to the results. We have clarified this in the main text (Sec 4, footnote). We also pointed out that future experiments will experiment with techniques proposed by  (Kotnis and Nastase 2017).
>
> Extra points:
>
> 1. Note that our goal was focusing on calibration, and not on achieving better predictive power. We have tried minimal random search on the hyperparameters without significant effect on triple classification results and we still could not reproduce the SOTA for FB13 and YAGO39K. Thus, we did not change the results of Table 4, besides adding the new HolE model. Some results in Table 4 incidentally achieve SOTA results, but we would rather leave the problem of achieving better predictive power aside.
>
> 1.1 Table 4 caption states that "for all calibration methods there is one single threshold \tau=0.5". We have added this to the header of Table 4 as well, for clarity. We added the per-relation decision thresholds in the appendix A.5. Note that the thresholds reported in A.5 are not probabilities, as they have been applied to the raw scores returned by the model-dependent scoring functions.
>
> 2. Fixed, thanks.
>
> 3. We made the figures more readable and printout friendly, as requested.

---

### Author Response · Authors · 2019-11-15
**Insightful reviews, thanks. Some general comments.**

Thank you for the comments. We have posted a personal reply to each reviewer. Let us also add a few general comments here:

* [Usefulness and importance of calibration]: Calibration will not impact rank-order metrics for link prediction, such as MRR. The usefulness of calibration lies on being able to trust the output of knowledge graph embedding models and even quantify this trust.
This has great importance in practice when discovering new links in biological networks: better calibrated probabilities help human experts (biologists) validating discoveries and make a ML pipeline based on graph embeddings more trustworthy.
As shown in the paper, an additional minor application is in the task of triple classification, where calibrated probabilities replace the need to learn arbitrary per-relation decision thresholds.

* [Novelty] We believe the paper is novel because it proposes the first framework for calibrating knowledge graph embedding models, something that to the best of our knowledge has never systematically been done before. It is certainly true that we rely on well-established calibration techniques, but we would like to point out that i) this is the first paper that covers this topic as first-class-citizen, and ii) the proposal of calibrating without positives is novel, as well as the use of sample weights to correct the distributional distortions created by the corruption generation procedure.


* We have made the following changes to the paper:

A) We expanded Table 3 results (impact of loss functions) in Sec 5.1 to include the other two datasets (FB13 and YAGO39K): We thank the reviewers for suggesting this additional experiment. In fact, we notice no correlation between calibration results and MRR. In other words, configurations that lead to the best predictive power are not necessary the best calibrated.

B) We have clarified in 5.1 that better or worse calibration has no impact on ranking metrics such as MRR. We only evaluate the hypothesis of embedding quality being a common cause of both MRR and calibration quality.

C) We added results and comments for two new experiments in appendix A.3, the impact of \eta and the embedding size on calibration: results show among all that the embedding size has higher impact than \eta.

D) We added the histograms that show the total count of instances for each bin used in the calibration plots. Figures and comment are in appendix A.2.

E) We have edited parts of the preliminary and related work, expanding the related work and condensing the preliminaries, as suggested. For example, now we mention KG2E (Gaussian embeddings) and many more additional recent papers.

F) We have fixed the typo on equation 6, thanks.

G) In the appendix, we added a table with MRR, MR, Hits@10 for all the datasets used. As pointed out in (B) above, we do not claim any causal relation of calibration on such task metrics.

H) Added per-relation decision thresholds in appendix A.5.

I) Move calibration-related preliminaries to appendix A.1.

J) All images are now black&white printout friendly.open

---

### Decision · Program_Chairs · 2019-12-19

**Decision:**

Accept (Poster)

**Comment:**

The paper proposes a novel method to calibrate a knowledge graph embedding method when ground truth negatives are not available. Essentially, the method relies on generating corrupted triples as negative examples to be used by known approaches (Platt scaling and isotonic regression).

This is claimed as the first approach of probability calibration for knowledge graph embedding models, which is considered to be very relevant for practitioners working on knowledge graph embedding (although this is a narrow audience). The paper does not propose a wholly novel method for probability calibration. Instead, the value in experimental insights provided.

Some reviewers would have liked to see a more in-depth analysis, but reviewers appreciated the thoroughness of the results in the clear articulation of the findings and the fact that multiple datasets and models are studied.

There was an animated discussion about this paper, but the paper seems a useful contribution to the ICLR community and I would like to recommend acceptance.